# AT-GAN: An Adversarial Generative Model for Non-constrained Adversarial Examples

## Abstract

With the rapid development of adversarial machine learning, numerous adversarial attack methods have been proposed. Typical attacks are based on a search in the neighborhood of input image to generate a perturbed adversarial example. Since 2017, generative models are adopted for adversarial attacks, and most of them focus on generating adversarial perturbations from input noise or input image. Thus the output is restricted by input for these works. A recent work targets "unrestricted adversarial example" using generative model but their method is based on a search in the neighborhood of input noise, so actually their output is still constrained by input. In this work, we propose AT-GAN (Adversarial Transfer on Generative Adversarial Net) to *train an adversarial generative model that can directly produce adversarial examples*. Different from previous works, we aim to *learn the distribution of adversarial examples* so as to generate semantically meaningful adversaries. AT-GAN achieves this goal by first learning a generative model for real data, followed by transfer learning to obtain the desired generative model. Once trained and transferred, AT-GAN could generate adversarial examples directly and quickly for any input noise, denoted as non-constrained adversarial examples. Extensive experiments and visualizations show that AT-GAN can efficiently generate diverse adversarial examples that are realistic to human perception, and yields higher attack success rates against adversarially trained models.

## 1 Introduction

In recent years, Deep Neural Networks (DNNs) have been found vulnerable to adversarial examples (Szegedy et al., 2014), which are well-crafted samples with tiny perturbations imperceptible to humans but can fool the learning models. Despite the great success of the deep learning empowered applications, many of them are safety-critical, for example under the scenario of self-driving cars (Eykholt et al., 2018; Cao et al., 2019), raising serious concerns in academy and industry.

Numerous works of adversarial examples have been developed on adversarial attacks (Goodfellow et al., 2015; Carlini & Wagner, 2017; Madry et al., 2018), adversarial defenses (Goodfellow et al., 2015; Kurakin et al., 2017; Song et al., 2019) and exploring the property of adversarial examples (He et al., 2018; Shamir et al., 2019). For adversarial attacks, most studies focus on the *perturbation-based adversarial examples* constrained by input images, which is also the generally accepted conception of adversarial examples. Generative models are also adopted recently to generate adversarial perturbations from an input noise (Reddy Mopuri et al., 2018; Omid et al., 2018) or from a given image (Xiao et al., 2018; Bai et al., 2020), and such perturbations are added to the original image to craft adversarial examples. Song et al. (2018) propose to search a neighborhood noise around the input noise of a Generative Adversarial Net (GAN) (Goodfellow et al., 2014) such that the output is an adversarial example, which they denoted as *unrestricted adversarial example* as there is no original image in their method. However, their output is still constrained by the input noise, and the search is time-consuming.

In this work, we propose an adversarial generative model called AT-GAN (Adversarial Transfer on Generative Adversarial Net), which aims to learn the distribution of adversarial examples. Unlike previous works that constrain the adversaries in the neighborhood of input image or input noise, including the prominent work of Song et al. (2018) that searches over the neighborhood of the input noise of a pre-trained GAN in order to find a noise whose output image is misclassified by the target classifier, AT-GAN is an adversarial generative model that could produce semantically meaningful

adversarial examples directly from any input noise, and we call such examples the *non-constrained adversarial examples*.

Specifically, we first develop a normal GAN to learn the distribution of benign data so that it can produce plausible images that the classifier and a human oracle will classify in the same way. Then we transfer the pre-trained GAN into an adversarial GAN called AT-GAN that can fool the target classifier while being still well recognized by the human oracle. AT-GAN is a conditional GAN that has learned to estimate the distribution of adversarial examples for the target classifier, so AT-GAN can directly generate adversarial examples from any random noise, leading to high diversity and efficiency.

We implement AT-GAN by adopting AC-GAN (Odena et al., 2017) and WGAN-GP (Gulrajani et al., 2017) in the pre-training stage, then do transfer learning for the adversary generation. Here we develop AT-GAN on three benchmark datasets, namely MNIST, Fashion-MNIST and CelebA, and apply typical defense methods to compare AT-GAN with existing search-based attacks. Empirical results show that the non-constrained adversarial examples generated by AT-GAN yield higher attack success rates, and state-of-the-art adversarially trained models exhibit little robustness against AT-GAN, indicating the high diversity of our adversaries. In addition, AT-GAN, as a generation-based adversarial attack, is more efficient than the search-based adversarial attacks.

Note that all conditional GANs that can craft realistic examples could be used for the implementation of AT-GAN. For another demonstration, we adopt StyleGAN2-ada (Karras et al., 2020a) and develop AT-GAN on CIFAR-10 benchmark dataset using wide ResNet w32-10 (Zagoruyko & Komodakis, 2016) as the target classifier. Empirical results show that AT-GAN can produce plausible adversarial images, and yield higher attack success rates on the adversarially trained models.

## 2 PRELIMINARIES

In this section, we provide definitions on several types of adversarial examples and adversarial attacks, and give a brief overview of adversarial attacks using GAN. Other related works on typical adversarial attacks and defenses (Goodfellow et al., 2015; Madry et al., 2018; Tramèr et al., 2018), as well as some typical GANs (Goodfellow et al., 2014; Radford et al., 2016; Odena et al., 2017; Arjovsky et al., 2017; Gulrajani et al., 2017) are introduced in Appendix A.

### 2.1 DEFINITIONS ON ADVERSARIES

Let $\mathcal{X}$ be the set of all digital images under consideration for a learning task, $\mathcal{Y} \in \mathbb{R}$ be the output label space and $p_z \in \mathbb{R}^m$ be an arbitrary probability distribution (*e.g.* Gaussian distribution) where $m$ is the dimension of $p_z$. A deep learning classifier $f : \mathcal{X} \to \mathcal{Y}$ takes an image $x \in \mathcal{X}$ and predicts its label $f(x)$. Suppose $p_x$ and $p_{adv}$ are the distributions of benign images and adversarial examples, respectively. Assume we have an oracle classifier $o : \mathcal{X} \to \mathcal{Y}$, which could always predict the correct label for any image $x \in \mathcal{X}$, we define several types of adversarial examples as follows.

For perturbation-based adversarial examples (Szegedy et al., 2014; Goodfellow et al., 2015; Moosavi-Dezfooli et al., 2016), tiny perturbations are added to the input images, which are imperceptible to humans but can cause the target classifier to make wrong predictions.

**Definition 1.** *Perturbation-based Adversarial Examples.* Given a subset (trainset or testset) images $\mathcal{T} \subset \mathcal{X}$ and a small constant $\epsilon > 0$, the perturbation-based adversarial examples can be defined as: $\mathcal{A}_p = \{x_{adv} \in \mathcal{X} | \exists x \in \mathcal{T}, \|x - x_{adv}\|_p < \epsilon \wedge f(x_{adv}) \neq o(x_{adv}) = f(x) = o(x)\}$.

Song et al. (2018) define a new type of adversarial examples called unrestricted adversarial examples, which is not related to the subset (trainset or testset) images, by adding adversarial perturbation to the input noise of a mapping, such as GAN, so that the output of the perturbed noise is an adversary to the target classifier.

**Definition 2.** *Unrestricted Adversarial Examples.* Given a mapping $G$ from $z \sim p_z$ to $G(z, y) \sim p_\theta$, where $p_\theta$ is an approximated distribution of $p_x$, and a small constant $\epsilon > 0$, the unrestricted adversarial examples can be defined as: $\mathcal{A}_u = \{G(z^*, y_s) \in \mathcal{X} | \exists z \sim p_z, z^* \sim p_z, \|z - z^*\|_p < \epsilon \wedge f(G(z^*, y_s)) \neq o(G(z^*, y_s)) = f(G(z, y_s)) = o(G(z, y_s)) = y_s\}$ where $y_s$ is the source label.

In this work, we train a conditional GAN to learn the distribution of adversarial examples and output the corresponding adversary directly from any input noise. To clarify the difference with Song et al. (2018), we call our generated adversaries the non-constrained adversarial examples.

**Definition 3.** *Non-constrained Adversarial Examples.* If there is a mapping $G^*$ from $z \sim p_z$ to $G^*(z, y) \sim q_\theta$, where $q_\theta$ is an approximated distribution of $p_{adv}$, the non-constrained adversarial examples can be defined as $\mathcal{A}_n = \{G^*(z, y_s) \in \mathcal{X} | f(G^*(z, y_s)) \neq o(G^*(z, y_s)) = y_s\}$ where $y_s$ is the source label.

Here we need to find a mapping $G^*$, *e.g.* a generative model, such that for $z \sim p_z$, $G^*(z, y)$ is an image in $\mathcal{X}$ and the output distribution is an approximated distribution of $p_{adv}$, for example using the Kullback-Leibler divergence (Kullback & Leibler, 1951), $KL(q_\theta || p_{adv}) < \epsilon$ for a small constant $\epsilon$.

In summary, *perturbation-based adversarial examples* are based on perturbing an image $x \in \mathcal{X}$, and *unrestricted adversarial examples* (Song et al., 2018) perturbs an input noise $z \sim p_z$ for an existing mapping $G$. Most perturbation-based adversarial attacks and Song et al. (2018) fall into the *search-based adversarial attack*.

**Definition 4.** *Search-based Adversarial Attack.* Given an input vector $v \in \mathcal{V}$ (either benign image $x$ or random vector $z$), the search-based adversarial attack searches a vector $v' : \|v - v'\|_p < \epsilon$ where $v'$ leads to an adversarial example for the target classifier.

In contrast, *non-constrained adversarial examples* are more generalized so that we need to learn a mapping $G^*$ such that for any input noise sampled from distribution $p_z$, the output is an adversarial image. Such a mapping to be learned is called an adversarial generative model, and our method falls into the *generation-based adversarial attack*.

**Definition 5.** *Generation-based Adversarial Attack.* Given an input vector $v \in \mathcal{V}$ (either benign image $x$ or random vector $z$), the generation-based adversarial attack generates adversarial perturbation or adversarial example directly from $v$, usually adopting generative models.

## 2.2 GENERATIVE MODELS FOR ADVERSARIAL ATTACK

Generative models have been adopted for adversarial attack in recent works (Baluja & Fischer, 2017). Reddy Mopuri et al. (2018) propose a Network for Adversary Generation (NAG) that models the distribution of adversarial perturbations for a target classifier so that their NAG can craft adversarial perturbations from any given random noise, which will be added to the natural image to fool the target classifier. Omid et al. (2018) propose to generate universal or image-dependent adversarial perturbations using U-Net (Ronneberger et al., 2015) or ResNet Generator (He et al., 2016) from any given random noise. Xiao et al. (2018) propose to train AdvGAN that takes an original image as the input and generate adversarial perturbation for the input to craft an adversarial example. Bai et al. (2020) further propose AI-GAN that adopts projected gradient descent (PGD) (Madry et al., 2018) in the training stage to train a GAN to generate target adversarial perturbation for the input image and target class. The above attack methods all fall into the generation-based adversarial attack, and their crafted examples fall into the perturbation-based adversarial examples. Another recent work called PS-GAN (Liu et al., 2019) pre-processes an input seed patch (a small image) to adversarial patch that will be added to a natural image to craft an adversarial example, and an attention model is used to locate the attack area on the natural image.

Different from the above methods that generate adversarial perturbations or patches, Song et al. (2018) propose to search a random noise $z^*$ around the input noise $z$ of AC-GAN (Odena et al., 2017) such that the corresponding output of AC-GAN is an adversarial example for the target classifier. Their method falls into the search-based adversarial attack, and their crafted examples fall into the unrestricted adversarial examples as there is no original image in their method.

AT-GAN falls into the generation-based adversarial attack, and the crafted examples fall into the non-constrained adversarial examples. To clearly distinguish our work, we highlight the differences with most related works as follows:

**NAG, AdvGAN and AI-GAN vs. AT-GAN.** NAG (Reddy Mopuri et al., 2018), AdvGAN (Xiao et al., 2018) and AI-GAN (Bai et al., 2020) focus on crafting adversarial perturbations by GANs. NAG takes random noise as input and crafts image-agnostic adversarial perturbation. AdvGAN and AI-GAN both use natural images as inputs, and generate the corresponding adversarial perturbations

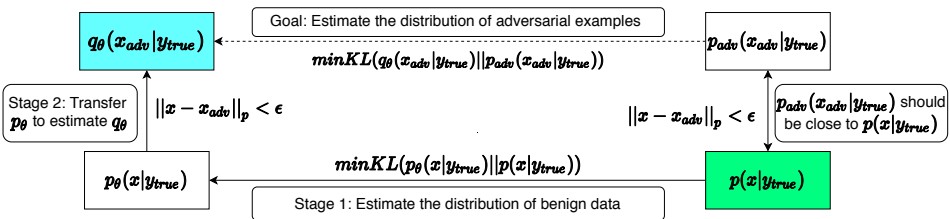

Figure 1: Estimating the distribution of adversarial examples $q_\theta$ in two stages: 1) estimate the distribution of benign data $p_\theta$. 2) transfer $p_\theta$ to estimate $q_\theta$.

for the input image. AI-GAN uses adversarial examples generated by PGD for the training. In contrast, AT-GAN does not use any natural image as the input, and generates adversarial examples directly from any random noise. Further, compared with AI-GAN, we do not use any adversarial examples for the training.

**Song's vs. AT-GAN.** Song's method (Song et al., 2018) searches over the neighborhood of the input noise for the pre-trained AC-GAN in order to find a noise whose output image is misclassified by the target classifier. They define such adversaries as the unrestricted adversarial examples, however, their adversaries are still constrained by the original input noise. Their method is essentially based on search, while AT-GAN is trained as an adversarial generative model, and our output is not constrained by any neighborhood.

## 3    AT-GAN: An Adversarial Generative Model

Here we first introduce the estimation on the distribution of adversarial examples, then propose the AT-GAN framework, a *generation-based adversarial attack* for crafting *non-constrained adversarial examples*. Further analysis is provided that AT-GAN could learn the adversary distribution.

### 3.1    Estimating the Adversarial Distribution

In order to generate non-constrained adversarial examples, we need to estimate the distribution of adversarial examples $p_{adv}(x_{adv}|y_{true})$ where $y_{true}$ is the true label. Given the parameterized estimated distribution of adversarial examples $q_\theta(x|y_{true})$, we can define the estimation problem as:

$$q_{\theta*}(x_{adv}|y_{true}) = \underset{\theta \in \Omega}{\arg\min}\, KL(q_\theta(x_{adv}|y_{true})\|p_{adv}(x_{adv}|y_{true})), \tag{1}$$

where $\theta$ indicates trainable parameters and $\Omega$ is the parameter space.

It is hard to calculate equation 1 directly as $p_{adv}(x_{adv}|y_{true})$ is unknown. Inspired by the perturbation-based adversarial examples, as shown in Figure 1, we postulate that for each adversarial example $x_{adv}$, there exists some benign examples $x$ where $\|x - x_{adv}\|_p < \epsilon$. In other words, $p_{adv}(x_{adv}|y_{true})$ is close to $p(x|y_{true})$ to some extent and we can obtain $p(x|y_{true})$ by Bayes' theorem, $p(x|y_{true}) = \frac{p(y_{true}|x) \cdot p(x)}{p(y_{true})}$, where $p(y_{true}|x)$, $p(x)$ and $p(y_{true})$ can be obtained directly from the trainset. Thus, we can approximately solve equation 1 in two stages: 1) Fit the distribution of benign data $p_\theta$. 2) Transfer $p_\theta$ to estimate the distribution of adversarial examples $q_\theta$.

Specifically, we propose an adversarial generative model called AT-GAN to learn the distribution of adversarial examples. The overall architecture of AT-GAN is illustrated in Figure 2. Corresponding to the above two stages, we implement AT-GAN by first training a GAN model called AC-WGAN_GP, which combines AC-GAN (Odena et al., 2017) and WGAN_GP (Gulrajani et al., 2017) to get a generator $G_{original}$, to learn $p_\theta$ (See Appendix B), then transfering $G_{original}$ to attack the target classifier $f$ for the learning of $q_\theta$. We adopt AC-GAN and WGAN-GP for the AT-GAN implementation as they could build a powerful generative model on three evaluated datasets, and Song et al. (2018) also utilize the same combination. But AT-GAN is not limited to the above GANs, and we also implement AT-GAN using StyleGAN2-ada (Karras et al., 2020a) on a different dataset.

### 3.2    Transferring the Generator for Attack

After the original generator $G_{original}$ is trained, we transfer the generator $G_{original}$ to learn the distribution of adversarial examples in order to attack the target model. As illustrated in Figure 2

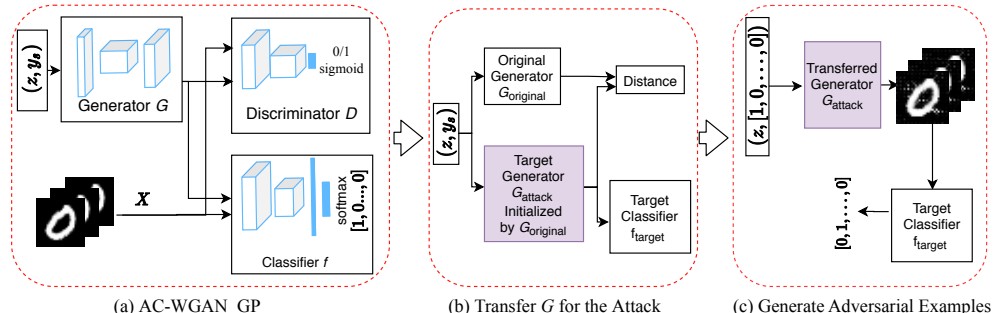

| (a) AC-WGAN_GP | (b) Transfer $G$ for the Attack | (c) Generate Adversarial Examples |

Figure 2: The architecture of AT-GAN. The first training stage of AT-GAN is same as AC-WGAN_GP. Once trained, we regard $G$ as the original model $G_{original}$ and transfer $G_{original}$ to attack target classifier to obtain $G_{attack}$. Once finished, AT-GAN can generate adversarial examples by $G_{attack}$.

(b), there are three neural networks, including the original generator $G_{original}$, the attack generator $G_{attack}$ to be transferred that is initialized by the weights of $G_{original}$, and the classifier $f$ to be attacked. The goal of the second stage can be described as:

$$G^*_{attack} = \underset{G_{attack}}{\arg\min} \; ||G_{original}(z, y_s) - G_{attack}(z, y_s)||_p \quad s.\,t.\; f(G(z, y_s)) = y_t \neq y_s, \quad (2)$$

where $y_t$ denotes the target label, $\| \cdot \|_p$ denotes the $\ell_p$ norm and we focus on $p = 2$ in this work.

To optimize equation 2, we construct the loss function by $L_1$ and $L_2$, where $L_1$ aims to assure that $f$ yields the target label $y_t$ that is fixed for target attack for each category:

$$L_1 = \mathbb{E}_{z \sim p_z}[H(f(G_{attack}(z, y_s)), y_t)]. \quad (3)$$

Here $H(\cdot, \cdot)$ denotes the cross entropy between the two terms and $y_s$ is sampled from $\mathcal{Y}$. $L_2$ aims to assure that the adversarial generator $G_{attack}$ generates realistic examples:

$$L_2 = \mathbb{E}_{z \sim p_z}[||G_{original}(z, y_s) + \rho - G_{attack}(z, y_s)||_p]. \quad (4)$$

Here $\rho$ is a small uniform random noise constrained by both $l_0$ and $l_\infty$ norm. We add $\rho$ to constrain $G_{attack}(z, y_s)$ to be in the neighborhood of $G_{original}(z, y_s)$ rather than be exactly the same as $G_{original}(z, y_s)$.

The objective function for transferring $G_{original}$ to $G_{attack}$ can be formulated as $L = \langle \alpha L_1, \beta L_2 \rangle$, where $\alpha$ and $\beta$ are hyper-parameters to control the training process. Note that in the case that $\alpha = 1$ and $\beta \to \infty$, the objective function is similar to that of the perturbation-based attacks (Goodfellow et al., 2015; Tramèr et al., 2018; Madry et al., 2018). For the untargeted attack, we can replace $y_t$ in $L_a$ with the maximum confidence of prediction label $y$ except for $y_s$, $\max_{y \neq y_s} f(y|G_{attack}(z, y_s))$.

### 3.3 THEORETICAL ANALYSIS ON AT-GAN

This subsection provides theoretical analysis on why AT-GAN can generate as realistic and diverse non-constrained adversarial examples as real data. We will prove that under ideal condition, AT-GAN can estimate the distribution of adversarial examples, which is close to that of real data.

Suppose $p_{data}$ is the distribution of real data, $p_g$ and $p_a$ are the distribution learned by the generator of AC-WGAN_GP and AT-GAN respectively. For the optimization of equation 4, $L_2$ aims to constrain the image generated by $G_{attack}$ in the $\epsilon$-neighborhood of $G_{original}$. We prove that under the ideal condition that $L_2$ guarantees $G_{attack}(z, y_s)$ to be close enough to $G_{original}(z, y_s)$ for any input noise $z$, the distribution of AT-GAN almost coincides the distribution of AC-WGAN_GP. Formally, we state our result for the two distributions as follows.

**Theorem 1.** Suppose $\max_{z,y} L_2 < \epsilon$, we have $KL(p_a \| p_g) \to 0$ when $\epsilon \to 0$.

The proof of Theorem 1 is in Appendix C. Samangouei et al. (2018) prove that the global optimum of WGAN is $p_g = p_{data}$ and we show that the optimum of AC-WGAN_GP has the same property. We formalize the property as follows.

**Theorem 2.** The global minimum of the virtual training of AC-WGAN_GP is achieved if and only if $p_g = p_{data}$.

The proof of Theorem 2 is in Appendix C. According to Theorem 1 and 2, under the ideal condition, we conclude $p_a \approx p_g = p_{data}$, which indicates that the distribution of non-constrained adversarial examples learned by AT-GAN is very close to that of real data as discussed in Section 3.1, so that the non-constrained adversarial instances are as realistic and diverse as the real data.

## 4 EXPERIMENTS

In this section, we provide two implementations of AT-GAN to validate the effectiveness and efficiency of the proposed approach. Empirical experiments demonstrate that AT-GAN yields higher attack success rates against adversarially trained models with higher efficiency. Besides, AT-GAN can learn a distribution of adversarial examples which is close to the real data distribution, and generate realistic and diverse adversarial examples.

### 4.1 EXPERIMENTAL SETUP

**Datasets.** We consider four standard datasets, namely MNIST (LeCun et al., 1989), Fashion-MNIST (Xiao et al., 2017), CelebA (Liu et al., 2015) on the AT-GAN implementation using AC-GAN (Odena et al., 2017) and WGAN_GP (Gulrajani et al., 2017), and CIFAR-10 dataset (Krizhevsky et al., 2009) on the AT-GAN implementation of StyleGAN2-ada (StyleGAN2 with adaptive discriminator augmentation) (Karras et al., 2020a). MNIST is a dataset of hand written digits from 0 to 9. Fashion-MNIST is similar to MNIST with 10 categories of fashion clothes. CelebA contains more than $200,000$ celebrity faces. We group them according to female/male and focus on gender classification as in Song et al. (2018). CIFAR-10 consists of $32 \times 32$ color images in 10 classes, with $6,000$ images per class. For all datasets, we normalize the pixel values into range $[0, 1]$.

**Baselines.** We compare AT-GAN with the search-based attack methods, including Song's (Song et al., 2018) for unrestricted adversarial examples, as well as FGSM (Goodfellow et al., 2015), PGD (Madry et al., 2018) and R+FGSM (Tramèr et al., 2018) for perturbation-based adversarial examples. Note that although the perturbation-based results are not directly comparable to ours as they are limited to small perturbations on real images, they can provide a good sense on the model robustness.

**Models.** For MNIST and Fashion-MNIST, we adopt four models used in Tramèr et al. (2018), denoted as Model A to D. For CelebA, we consider three models, *i.e.* CNN, VGG16 (Simonyan & Zisserman, 2015) and ResNet (He et al., 2016). Details of Model A to D and CNN are described in Table 1. The ResNet is same as in Song et al. (2018). For CIFAR-10, we adopt the wide ResNet w32-10 (Zagoruyko & Komodakis, 2016). Details about the architectures of AT-GAN are provided in Appendix D.

Table 1: Architectures of Model A through D used for MNIST and Fashion-MNIST and CNN used for CelebA. The total number of parameters of each model is provided after the model name.

| Model A $(3,382,346)$ | Model B $(710,218)$ | Model C $(4,795,082)$ | Model D $(509,410)$ | CNN $(17,066,658)$ |
|---|---|---|---|---|
| Conv$(64, 5 \times 5)$+Relu | Dropout$(0.2)$ | Conv$(128, 3 \times 3)$+Relu | $\begin{bmatrix} \text{FC}(300)\text{+Relu} \\ \text{Dropout}(0.5) \end{bmatrix} \times 4$ | [Conv$(32, 3 \times 3)$+Relu] $\times 2$ |
| Conv$(64, 5 \times 5)$+Relu | Conv$(64, 8 \times 8)$+Relu | Conv$(64, 3 \times 3)$+Relu | | Dropout$(0.3)$ |
| Dropout$(0.25)$ | Conv$(128, 6 \times 6)$+Relu | Dropout$(0.25)$ | | [Conv$(64, 3 \times 3)$+Relu] $\times 2$ |
| FC$(128)$+Relu | Conv$(128, 5 \times 5)$+Relu | FC$(128)$+Relu | FC$(10)$ + Softmax | Maxpool$(2, 2)$ + Dropout$(0.3)$ |
| Dropout$(0.5)$ | Dropout$(0.5)$ | Droopout$(0.5)$ | | [Conv$(128, 3 \times 3)$+Relu]$\times 2$ |
| FC$(10)$+Softmax | FC$(10)$+Softmax | FC$(10)$+Softmax | | Maxpool$(2, 2)$ + Dropout$(0.3)$ |
| | | | | FC$(512)$ + Relu |
| | | | | Dropout$(0.3)$ |
| | | | | FC$(10)$ + Softmax |

**Evaluation Setup.** We consider normal training and existing advanced defenses, namely adversarial training (Goodfellow et al., 2015), ensemble adversarial training (Tramèr et al., 2018) and iterative adversarial training (Madry et al., 2018). All experiments are conducted on a single Titan X GPU and the hyper-parameters used for attacks are described in Appendix D.

### 4.2 EVALUATION RESULTS

For evaluation, we report the comparisons on attack success rate, attack efficiency and visualize some adversarial examples for AT-GAN and the baselines. More evaluation results on the transferability, ablation study, human evaluation, and the attack results on CIFAR-10, are provided in Appendix D.

Table 2: Attack success rate (ASR, %) of adversarial examples generated by AT-GAN and the baseline attacks against models by normal training and various adversarial training methods. For each model, the highest ASR is highlighted in **bold**. Notation: Nor. — Normal training, Adv. — Adversarial training, Ens. — Ensemble adversarial training, Iter. — Iterative adversarial training.

(a) Comparison of attack success rate on MNIST.

| Attack | Model A | | | | Model B | | | | Model C | | | | Model D | | | |
|---|---|---|---|---|---|---|---|---|---|---|---|---|---|---|---|---|
| | Nor. | Adv. | Ens. | Iter. | Nor. | Adv. | Ens. | Iter. | Nor. | Adv. | Ens. | Iter. | Nor. | Adv. | Ens. | Iter. |
| FGSM | 68.8 | 2.9 | 8.7 | 4.0 | 88.0 | 9.5 | 18.0 | 9.2 | 70.7 | 4.3 | 7.8 | 4.7 | 89.6 | 23.5 | 34.6 | 26.7 |
| PGD | **100.0** | 92.6 | 85.1 | 5.9 | **100.0** | 42.4 | 98.2 | 36.2 | **100.0** | 76.9 | 96.4 | 9.6 | 91.7 | 96.8 | **99.5** | 81.1 |
| R+FGSM | 77.9 | 28.4 | 20.3 | 2.6 | 96.5 | 7.1 | 42.1 | 4.6 | 81.6 | 7.2 | 19.9 | 2.9 | 93.8 | 76.2 | 51.6 | 25.3 |
| Song's | 82.0 | 70.5 | 75.0 | 84.6 | 76.9 | 65.0 | 72.0 | 80.7 | 74.2 | 75.6 | 72.6 | 87.8 | 67.7 | 43.6 | 56.3 | 44.5 |
| AT-GAN | 98.7 | **97.5** | **96.7** | **91.4** | 99.5 | **97.7** | **99.3** | **95.6** | 99.3 | **95.8** | **96.9** | **90.0** | **99.9** | **99.9** | 99.5 | **99.7** |

(b) Comparison of attack success rate on Fashion-MNIST.

| Attack | Model A | | | | Model B | | | | Model C | | | | Model D | | | |
|---|---|---|---|---|---|---|---|---|---|---|---|---|---|---|---|---|
| | Nor. | Adv. | Ens. | Iter. | Nor. | Adv. | Ens. | Iter. | Nor. | Adv. | Ens. | Iter. | Nor. | Adv. | Ens. | Iter. |
| FGSM | 68.8 | 2.9 | 8.7 | 4.0 | 88.0 | 9.5 | 18.0 | 9.2 | 70.7 | 4.3 | 7.8 | 4.7 | 89.6 | 23.5 | 34.6 | 26.7 |
| PGD | **100.0** | 92.6 | 85.1 | 5.9 | **100.0** | 42.4 | 98.2 | 36.2 | **100.0** | 76.9 | 96.4 | 9.6 | 91.7 | 96.8 | **99.5** | 81.1 |
| R+FGSM | 77.9 | 28.4 | 20.3 | 2.6 | 96.5 | 7.1 | 42.1 | 4.6 | 81.6 | 7.2 | 19.9 | 2.9 | 93.8 | 76.2 | 51.6 | 25.3 |
| Song's | 82.0 | 70.5 | 75.0 | 84.6 | 76.9 | 65.0 | 72.0 | 80.7 | 74.2 | 75.6 | 72.6 | 87.8 | 67.7 | 43.6 | 56.3 | 44.5 |
| AT-GAN | 98.7 | **97.5** | **96.7** | **91.4** | 99.5 | **97.7** | **99.3** | **95.6** | 99.3 | **95.8** | **96.9** | **90.0** | **99.9** | **99.9** | 99.5 | **99.7** |

(c) Comparison of attack success rate on CelebA.

| Attack | CNN | | | | VGG16 | | | | ResNet | | | |
|---|---|---|---|---|---|---|---|---|---|---|---|---|
| | Nor. | Adv. | Ens. | Iter. | Nor. | Adv. | Ens. | Iter. | Nor. | Adv. | Ens. | Iter. |
| FGSM | 81.2 | 11.8 | 14.7 | 9.5 | 76.7 | 10.6 | 16.2 | 8.9 | 98.7 | 9.8 | 12.3 | 9.5 |
| PGD | 97.3 | 16.4 | 22.6 | 11.4 | 87.9 | 14.6 | 26.3 | 10.7 | **100.0** | 10.9 | 15.1 | 10.5 |
| R+FGSM | 68.7 | 9.5 | 11.3 | 7.9 | 68.4 | 8.7 | 13.2 | 7.3 | 97.5 | 7.9 | 9.5 | 7.8 |
| Song's | 90.6 | 83.4 | 85.7 | 89.8 | **98.7** | 87.5 | 95.7 | 81.6 | 99.2 | 93.4 | 91.0 | 90.6 |
| AT-GAN | 97.5 | **98.9** | **95.9** | **99.6** | 97.1 | **96.7** | **95.8** | **97.8** | 98.4 | **98.5** | **97.3** | **98.5** |

### 4.2.1 COMPARISON ON ATTACK SUCCESS RATE

To validate the attack effectiveness, we compare AT-GAN with the baselines under white-box setting. Since Athalye et al. (2018) show that the currently most effective defense method is adversarial training, we consider adversarially trained models as the defense models. The attack success rates are reported in Table 2.

On MNIST, AT-GAN achieves the highest Attack Success Rate (ASR) against the baselines on all defense models. As for normal training, AT-GAN achieves the highest ASR on Model D, and the second highest ASR of over $98\%$ on the other models. On Fashion-MNIST, AT-GAN achieves the highest ASR on average. On CelebA, AT-GAN achieves the highest ASR on almost all the models, with two exceptions under normal training but the results of AT-GAN are close to the highest.

In general, AT-GAN achieves the highest attack performance above 90% on all the defense models. *As AT-GAN aims to estimate the distribution of adversarial examples, adversarial training with some specific attacks has little robustness against AT-GAN, raising a new security issue for the development of more generalized adversarial training models.*

### 4.2.2 COMPARISON ON ATTACK EFFICIENCY

There are many scenarios where one needs a large amount of adversarial examples, such as adversarial training or exploring the property of adversarial examples. Thus, the efficiency of generating adversarial examples is very important, but such metric is ignored in most existing works.

As an adversarial generative model, once trained, AT-GAN can generate adversarial examples very quickly. Here we evaluate the efficiency of each attack method for Model A on MNIST. The average time of generating/searching 1000 adversarial examples is summarized in Table 3. Among the five attack methods, AT-GAN is the fastest as it could craft adversarial examples without target classifier and gradient calculation. Note that Song's needs much longer time than others as it needs multiple searches and queries to generate one adversarial example. It takes about 8 minutes for transferring the generator of AT-GAN. Here we only focus on the efficiency of generating adversarial examples after AT-GAN is transferred, *i.e.* we have already found the generator $G^*$, as in such case we could generate as many adversarial examples as we need.

Table 3: Comparison on the average example generation time, measured by generating 1000 adversarial instances using Model A on MNIST.

|  | FGSM | PGD | R+FGSM | Song's | AT-GAN |
|---|---|---|---|---|---|
| Time | 0.3s | 1.8s | 0.4s | $\geq 2.5min$ | 0.2s |

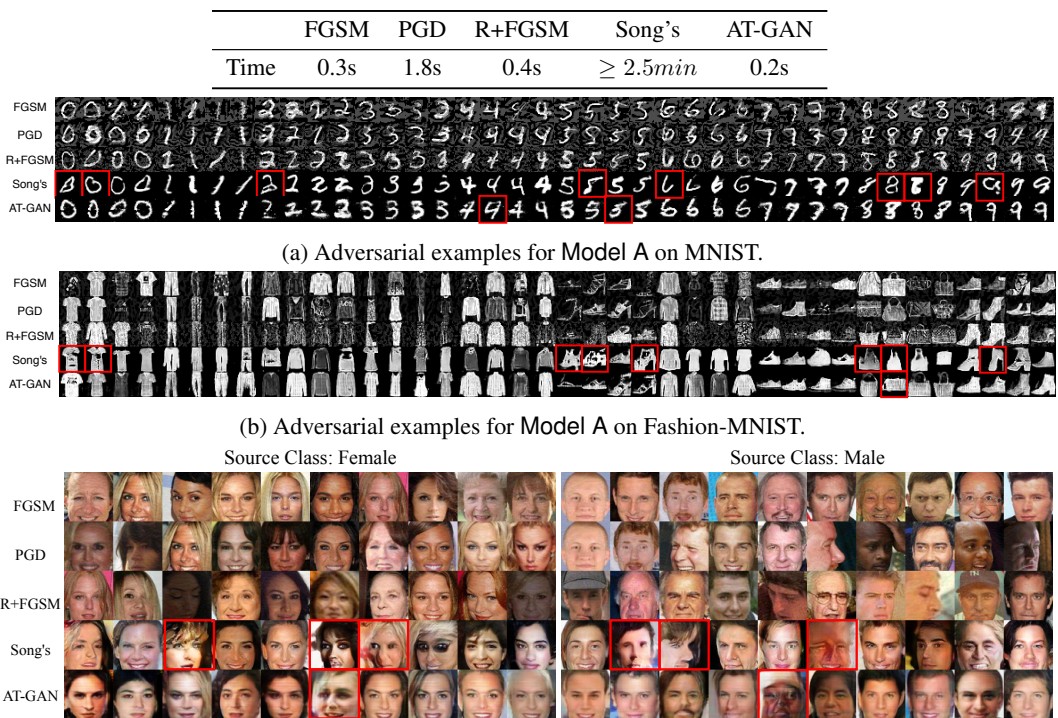

(a) Adversarial examples for Model A on MNIST.

(b) Adversarial examples for Model A on Fashion-MNIST.

(c) Adversarial examples for CNN on CelebA.

Figure 3: Adversarial examples generated by various attacks on three datasets (Zoom in for details). The red borders indicate unrealistic adversarial examples generated by Song's method or AT-GAN.

### 4.2.3 VISUALIZATION ON ADVERSARIAL EXAMPLES

Since the goal of adversarial examples is to fool target neural networks but not to fool human oracle, in Figure 3 we illustrate some adversarial examples generated by different attacks for Modle A on MNIST and Fashion-MNIST, and CNN on CelebA.

On MNIST, AT-GAN generates slightly more realistic images than Song's, *e.g.* "0" and "3". On Fashion-MNIST and CelebA, some adversarial examples generated by Song's method are not as realistic as AT-GAN to human perception, for example "t-shirt/top (0) ", "sandal (5)" and some facial details. Note that Song's method tends to distort the foreground that makes the images on MNIST more clean but some images are not realistic while AT-GAN tends to distort the background. As for perturbation-based attacks, their adversarial examples are not clear enough, especially on MNIST and Fashion-MNIST, due to the adversarial perturbations. There are also some unnatural samples generated by AT-GAN due to the limitation of GAN and we hope some better generative models can solve such issue. For target attack, please see more examples crafted by AT-GAN in Appendix D.

In general, AT-GAN can generate realistic and diverse adversarial examples as equation 1 forces the generated non-constrained adversarial examples to be close to the benign examples generated by the original generator.

### 4.3 VISUALIZATION ON ADVERSARIAL DISTRIBUTION

As discussed in Section 3.3, we provide a brief analysis that AT-GAN can learn a distribution of adversarial examples close to the distribution of real image data. To identify it empirically, we randomly choose $5,000$ benign images and $5,000$ adversarial examples generated by different attack methods, and merge these images according to their real label for MNIST and Fashion-MNIST. Then we use t-SNE (Maaten & Hinton, 2008) on these images to illustrate the distributions in two dimensions. t-SNE models each high-dimensional object in such a way that similar objects are

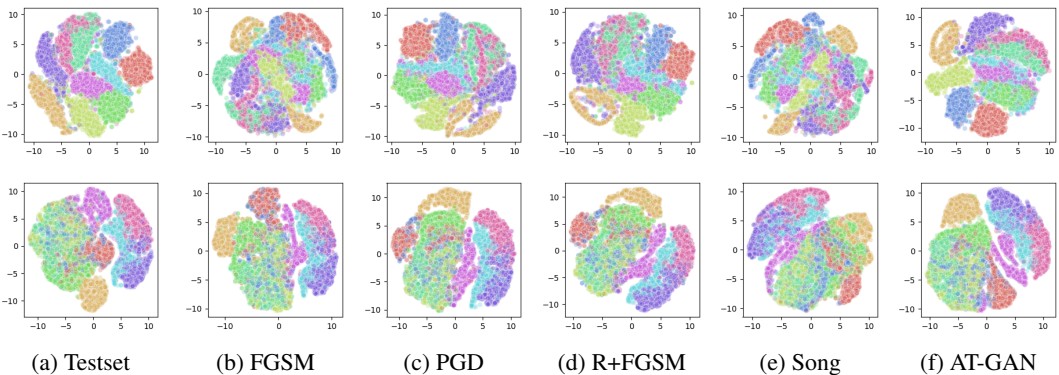

|        (a) Testset | (b) FGSM | (c) PGD | (d) R+FGSM | (e) Song | (f) AT-GAN |

Figure 4: T-SNE visualization for the combination of testset and adversarial examples generated by various attacks on MNIST (top) and Fashion-MNIST (bottom). For (a), we use total testset with 10,000 images. For (b) to (f), we use 5,000 sampled images in testset and 5,000 adversarial examples generated by various attacks. The position of each class is random due to the property of t-SNE.

modeled by nearby points and dissimilar objects are modeled by distant points with high probability. It indicates that, if the adversarial examples have different distribution to the benign data, t-SNE could not deal with them well and the points with different categories will overlap with each other after the dimension reduction, *i.e.* the results will be in chaos.

The results are illustrated in Figure 4. For AT-GAN, different categories are separated as that of the test set while those of other methods are mixed with each other, especially on MNIST (top). It indicates the distribution AT-GAN learned is indeed very close to the distribution of real data.

To further validate that AT-GAN learns a different distribution from the original GAN rather than just adding some constant universal perturbation vector. In Appendix E, we illustrate some instances generated by the original generator and AT-GAN for the same input. We find that for different inputs, the original generator outputs different images and the difference between the instances generated by the original generator and AT-GAN is also different, indicating that AT-GAN indeed learns a different distribution from the original GAN.

## 5 CONCLUSION

In this work, we propose a generation-based adversarial attack method, called AT-GAN (Adversarial Transfer on Generative Adversarial Net), that aims to learn the distribution of adversarial examples for the target classifier. The generated adversaries are "non-constrained" as we do no search at all in the neighborhood of the input, and once trained AT-GAN can output adversarial examples directly for any input noise drawn from arbitrary distribution (e.g. Gaussian distribution). Extensive experiments and visualizations show that AT-GAN achieves highest attack success rates against adversarially trained models and can generate diverse and realistic adversarial examples efficiently.

Our work also suggests that adversarial training, a popular defense method based on perturbation-based adversarial examples, could not guarantee robustness against non-constrained adversarial examples. A possible reason is that AT-GAN learns a more complete version of the adversarial example distribution, which is much more diverse than that of the perturbation-based method.

Note that any conditional GANs that craft realistic examples could be used for the implementation of AT-GAN. In this work, we provide two implementations on four datasets. In future work we plan to try advanced GANs for generating high resolution images. Our method also suggests a new way of adversarial attack by designing an adversarial generative model directly. There are several other interesting questions related to our work that can be explored in future work. For instance, what is the distribution of adversarial examples really like? Is it a continuous or smooth manifold? How close could we learn such distribution through GAN? We hope our work could inspire more researches in this direction.

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

APPENDIX

In the appendix, we provide additional related work on gradient-based adversarial attack methods, adversarial training methods and typical generative adversarial nets. Then we describe how to obtain the original generator and provide theoretical analysis, as well as experimental details and additional results. In the end, we visualize the examples generated by original GAN and AT-GAN.

## A  ADDITIONAL RELATED WORK

### A.1  GRADIENT-BASED ATTACKS

Numerous adversarial attacks have been proposed in recent years (Carlini & Wagner, 2017; Liu et al., 2017; Bhagoji et al., 2017; Li et al., 2019). In this part, we will introduce three typical adversarial attack methods. Here the components of all adversarial examples are clipped in $[0, 1]$.

**Fast Gradient Sign Method (FGSM).** FGSM (Goodfellow et al., 2015) adds perturbation in the gradient direction of the training loss $J$ on the input $x$ to generate adversarial examples.

$$x_{adv} = x + \epsilon \cdot sign(\nabla_x J(\theta, x, y_{true})),$$

where $y_{true}$ is the true label of a sample $x$, $\theta$ is the model parameter and $\epsilon$ specifies the $\ell_\infty$ distortion between $x$ and $x_{adv}$.

**Projected Gradient Descent (PGD).** PGD adversary (Madry et al., 2018) is a multi-step variant of FGSM, which applies FGSM for $k$ iterations with a budget $\alpha$.

$$x_{adv_{t+1}} = \mathbf{clip}(x_{adv_t} + \alpha sign(\nabla_x J(\theta, x_{adv_t}, y_{true})), x_{adv_t} - \epsilon, x_{adv_t} + \epsilon)$$
$$x_{adv_0} = x, \quad x_{adv} = x_{adv_k}$$

Here $\mathbf{clip}(x', p, q)$ forces its input $x'$ to reside in the range of $[p, q]$.

**Rand FGSM (R+FGSM).** R+FGSM (Tramèr et al., 2018) first applies a small random perturbation on the benign image with a parameter $\alpha$ ($\alpha < \epsilon$), then it uses FGSM to generate an adversarial example based on the perturbed image.

$$x_{adv} = x' + (\epsilon - \alpha) \cdot sign(\nabla_{x'} J(\theta, x', y_{true})) \quad \text{where } x' = x + \alpha \cdot sign(\mathcal{N}(\mathbf{0}, \mathbf{I})).$$

### A.2  ADVERSARIAL TRAINING

There are many defense strategies, such as detecting adversarial perturbations (Metzen et al., 2017), obfuscating gradients (Buckman et al., 2018; Guo et al., 2018) and eliminating perturbations (Shen et al., 2017; Liao et al., 2018), among which adversarial training is the most effective method (Athalye et al., 2018). We list several adversarial training methods as follows.

**Adversarial training.** Goodfellow et al. (2015) first introduce the method of adversarial training, where the standard loss function $f$ for a neural network is modified as:

$$\tilde{J}(\theta, x, y_{true}) = \alpha J_f(\theta, x, y_{true}) + (1 - \alpha)J_f(\theta, x_{adv}, y_{true}).$$

Here $y_{true}$ is the true label of a sample $x$ and $\theta$ is the model's parameter. The modified objective is to make the neural network more robust by penalizing it to count for adversarial samples. During the training, the adversarial samples are calculated with respect to the current status of the network. Taking FGSM for example, the loss function could be written as:

$$\tilde{J}(\theta, x, y_{true}) = \alpha J_f(\theta, x, y_{true}) + (1 - \alpha)J_f(\theta, x + \epsilon sign(\nabla_x J(\theta, x, y_{true})), y_{true}).$$

**Ensemble adversarial training.** Tramèr et al. (2018) propose an ensemble adversarial training method, in which DNN is trained with adversarial examples transferred from a number of fixed pre-trained models.

**Iterative adversarial training.** Madry et al. (2018) propose to train a DNN with adversarial examples generated by iterative methods such as PGD.

### A.3 GENERATIVE ADVERSARIAL NET

Generative Adversarial Net (GAN) (Goodfellow et al., 2014) consists of two neural networks, $G$ and $D$, trained in opposition to each other. The generator $G$ is optimized to estimate the data distribution and the discriminator $D$ aims to distinguish fake samples from $G$ and real samples from the training data. The objective of $D$ and $G$ can be formalized as a min-max value function $V(G, D)$:

$$\min_G \max_D V(G, D) = \mathbb{E}_{x \sim p_x}[\log D(x)] + \mathbb{E}_{z \sim p_z}[\log(1 - D(G(z)))].$$

Deep Convolutional Generative Adversarial Net (DCGAN) (Radford et al., 2016) is the convolutional version of GAN, which implements GAN with convolutional networks and stabilizes the training process. Auxiliary Classifier GAN (AC-GAN) (Odena et al., 2017) is another variant that extends GAN with some conditions by an extra classifier $C$. The objective function of AC-GAN can be formalized as follows:

$$\min_G \max_D \min_C V(G, D, C) = \mathbb{E}_{x \sim p_x}[\log D(x)] + \mathbb{E}_{z \sim p_z}[\log(1 - D(G(z, y_s)))] \\ + \mathbb{E}_{x \sim p_x}[\log(1 - C(x, y_s))] + \mathbb{E}_{z \sim p_z}[\log(1 - C(G(z, y_s), y_s))].$$

To make GAN more trainable in practice, Arjovsky et al. (2017) propose Wasserstein GAN (WGAN) that uses Wassertein distance so that the loss function has more desirable properties. Gulrajani et al. (2017) introduce WGAN with gradient penalty (WGAN_GP) that outperforms WGAN in practice. Its objective function is formulated as:

$$\min_G \max_D V(D, G) = \mathbb{E}_{x \sim p_x}[D(x)] - \mathbb{E}_{z \sim p_z}[D(G(z))] - \lambda \mathbb{E}_{\hat{x} \sim p_{\hat{x}}}[(\|\nabla_{\hat{x}} D(\hat{x})\|_2 - 1)^2],$$

where $p_{\hat{x}}$ is uniformly sampled along straight lines between pairs of points sampled from the data distribution $p_x$ and the generator distribution $p_g$.

## B TRAINING THE ORIGINAL GENERATOR

Figure 2 (a) illustrates the overall architecture of AC-WGAN_GP that we used as the normal GAN. AC-WGAN_GP is the combination of AC-GAN (Odena et al., 2017) and WGAN_GP (Gulrajani et al., 2017), composed by three neural networks: a generator $G$, a discriminator $D$ and a classifier $f$. The generator $G$ takes a random noise $z$ and a source label $y_s$ as the inputs and generates an image $G(z, y_s)$. It aims to generate an image $G(z, y_s)$ that is indistinguishable to discriminator $D$ and makes the classifier $f$ to output label $y_s$. The loss function of $G$ can be formulated as:

$$L_G = \mathbb{E}_{z \sim p_z(z)}[H(f(G(z, y_s)), y_s)] - \mathbb{E}_{z \sim p_z(z)}[D(G(z, y_s))].$$

Here $H(a, b)$ is the entropy between $a$ and $b$. The discriminator $D$ takes the training data $x$ or the generated data $G(z, y_s)$ as the input and tries to distinguish them. The loss function of $D$ with gradient penalty for samples $\hat{x} \sim p_{\hat{x}}$ can be formulated as:

$$L_D = -\mathbb{E}_{x \sim p_{data}(x)}[D(x)] + \mathbb{E}_{z \sim p_z(z)}[D(G(z, y_s))] + \lambda \mathbb{E}_{\hat{x} \sim p_{\hat{x}}(\hat{x})}[(\|\nabla_{\hat{x}} D(\hat{x})\|_2 - 1)^2].$$

The classifier $f$ takes the training data $x$ or the generated data $G(z, y_s)$ as the input and predicts the corresponding label. The loss function is:

$$L_f = \mathbb{E}_{x \sim p_{data}(x)}[H(f(x), y_{true})] + \mathbb{E}_{z \sim p_z(z)}[H(f(G(z, y_s)), y_s)].$$

Different from AC-WGAN_GP, styleGAN2-ada (Karras et al., 2020a) trains styleGAN2 (Karras et al., 2020b) with adaptive discriminator augmentation. We obtain the network and weights from Karras et al. (2020a).

## C THEORETICAL ANALYSIS OF AT-GAN

In this section, we provide proofs for theorems in Section 3.3.

**Theorem 1.** Suppose $\max_{z,y} L_2 < \epsilon$, we have $KL(p_a \| p_g) \to 0$ when $\epsilon \to 0$.

*Proof.* We first consider that for a distribution $p(x)$ in space $\mathcal{X}$, we construct another distribution $q(x)$ by selecting points $p_\epsilon(x)$ in the $\epsilon$-neighborhood of $p(x)$ for any $x \in \mathcal{X}$. Obviously, when $p_\epsilon(x)$ is close enough to $p(x)$, $q(x)$ has almost the same distribution as $p(x)$. Formally, we have the following lemma.

**Lemma 1.** Given two distributions $P$ and $Q$ with probability density function $p(x)$ and $q(x)$ in space $\mathcal{X}$, if there exists a constant $\epsilon$ that satisfies $\|q(x) - p(x)\| < \epsilon$ for any $x \in \mathcal{X}$, we could get $KL(P\|Q) \to 0$ when $\epsilon \to 0$.

*Proof.* For two distributions $P$ and $Q$ with probability density function $p(x)$ and $q(x)$, we could get $q(x) = p(x) + r(x)$ where $\|r(x)\| < \epsilon$.

$$
\begin{aligned}
KL(P\|Q) &= \int p(x) \log \frac{p(x)}{q(x)} dx \\
&= \int p(x) \log p(x) dx - \int p(x) \log q(x) dx \\
&= \int (q(x) - r(x)) \log p(x) dx - \int (q(x) - r(x)) \log q(x) dx \\
&= \int q(x) \log p(x) dx - \int q(x) \log q(x) dx - \int r(x) \log p(x) dx + \int r(x) \log q(x) dx \\
&= \int r(x) \log \frac{q(x)}{p(x)} dx - KL(Q\|P) \\
&\leq \int \epsilon \log(1 + \frac{\epsilon}{p(x)}) dx
\end{aligned}
$$

Obviously, when $\epsilon \to 0$, we could get $\int \epsilon \log(1 + \frac{\epsilon}{p(x)}) dx \to 0$, which means $DL(P\|Q) \to 0$. □

Now, we get back to Theorem 1. For two distributions $p_a$ and $p_g$, $\max_{y,z} L_2 < \epsilon$ indicates $\forall z \sim p_z, \|p_a(z, \cdot) - p_g(z, \cdot)\| < \epsilon$. According to Lemma 1, we have $KL(p_a\|p_g) \to 0$ when $\epsilon \to 0$. This concludes the proof. □

**Theorem 2.** The global minimum of the virtual training of AC-WGAN_GP is achieved if and only if $p_g = p_{data}$.

*Proof.* To simplify the analysis, we choose a category $y$ of AC-WGAN_GP and denote $p_g(x|y)$ and $p_{data}(x|y)$ the distribution that the generator learns and the distribution of real data respectively. Then for each category, the loss function is equivalent to WGAN_GP. We refers to Samangouei et al. (2018) to prove this property. The WGAN_GP min-max loss is given by:

$$
\begin{aligned}
\min_G \max_D V(D, G) &= \mathbb{E}_{x \sim p_{data}(x)}[D(x)] - \mathbb{E}_{z \sim p_z(z)}[D(G(z))] - \lambda \mathbb{E}_{\hat{x} \sim p_{\hat{x}}(\hat{x})}[(\|\nabla_{\hat{x}} D(\hat{x})\|_2 - 1)^2] \\
&= \int_x p_{data}(x) D(x) dx - \int_z p_z(z) D(G(z)) dz - \lambda \int_{\hat{x}} p_{\hat{x}}(\hat{x})[(\|\nabla_{\hat{x}} D(\hat{x})\|_2 - 1)^2] d\hat{x} \\
&= \int_x [p_{data}(x) - p_g(x)] D(x) dx - \lambda \int_{\hat{x}} p_{\hat{x}}(\hat{x})[(\|\nabla_{\hat{x}} D(\hat{x})\|_2 - 1)^2] d\hat{x}
\end{aligned}
$$

(5)

For a fixed $G$, the optimal discriminator $D$ that maximizes $V(D, G)$ should be:

$$
D_G^*(x) = \begin{cases} 1 & \text{if } p_{data}(x) \geq p_g(x) \\ 0 & \text{otherwise} \end{cases}
$$

(6)

According to equation 5 and equation 6, we could get:

$$
\begin{aligned}
V(D, G) &= \int_x [p_{data}(x) - p_g(x)] D(x) dx - \lambda \int_{\hat{x}} p_{\hat{x}}(\hat{x})[(\|\nabla_{\hat{x}} D(\hat{x})\|_2 - 1)^2] d\hat{x} \\
&= \int_{\{x|p_{data}(x) \geq p_g(x)\}} (p_{data}(x) - p_g(x)) dx - \lambda \int_{\hat{x}} p_{\hat{x}}(\hat{x}) d\hat{x} \\
&= \int_{\{x|p_{data}(x) \geq p_g(x)\}} (p_{data}(x) - p_g(x)) dx - \lambda
\end{aligned}
$$

(7)

Let $\mathcal{X} = \{x|p_{data}(x) \geq p_g(x)\}$, in order to minimize equation 7, we set $p_{data}(x) = p_g(x)$ for any $x \in \mathcal{X}$. Then, since both $p_g$ and $p_{data}$ integrate to 1, we could get:

$$\int_{\mathcal{X}^c} p_g(x)dx = \int_{\mathcal{X}^c} p_{data}(x)dx.$$

However, this contradicts equation 6 where $p_{data}(x) < p_g(x)$ for $x \in \mathcal{X}^c$, unless $\mu(\mathcal{X}^c) = 0$ where $\mu$ is the Lebesgue measure.

Therefore, for each category we have $p_g(x|y) = p_{data}(x|y)$, which means $p_g(x) = p_{data}(x)$ for AC-WGAN_GP. □

## D   ADDITIONAL DETAILS ON EXPERIMENTS

In this section, we provide more details on experimental setup, report results on transferability, do ablation study on hyper-parameters, investigate the generating capacity by human evaluation, and show details for another implementation of AT-GAN on CIFAR-10 dataset. In the end, we illustrate some non-constrained adversarial examples generated by AT-GAN on MNIST, Fashion-MNIST and CelebA for the target attack.

### D.1   MORE EXPERIMENTAL SETUP

We first provide more details on the experimental setup, including the model architectures and attack hyper-parameters.

**Model Architectures for AT-GAN.** We first describe the neural network architectures used for AT-GAN in experiments. The abbreviations for components in the network are described in Table 4. The architecture of AC-WGAN_GP for MNIST and Fashion-MNIST is shown in Table 5 where the generator and discriminator are the same as in Chen et al. (2016), while the architecture of AC_WGAN_GP for CelebA is the same as in Gulrajani et al. (2017) and the architecture of styleGAN2-ada for CIFAR-10 is the same as in Karras et al. (2020a).

**Hyper-parameters for Attacks.** The hyper-parameters used in experiments for each attack method are described in Table 6 for MNIST, Fashion-MNIST and CelebA datasets. For CIFAR-10 dataset, we set $\epsilon = 0.03$ for FGSM, $\epsilon = 0.03$, $\alpha = 0.0075$ and epochs$= 20$ for PGD, $\alpha = 3$, $\beta = 2$ and epochs$= 1,000$ for AT-GAN.

Table 4: Abbreviations for network architectures.

| Abbreviation | Description |
|---|---|
| Conv($m$, $k \times k$) | A convolutional layer with $m$ filters and filter size $k$ |
| DeConv($m$, $k \times k$) | A transposed convolutional layer with $m$ filters and filter size $k$ |
| Dropout($\alpha$) | A dropout layer with probability $\alpha$ |
| FC($m$) | A fully connected layer with $m$ outputs |
| Sigmoid | The sigmoid activation function |
| Relu | The Rectified Linear Unit activation function |
| LeakyRelu($\alpha$) | The Leaky version of a Rectified Linear Unit with parameter $\alpha$ |
| Maxpool($k$,$s$) | The maxpooling with filter size $k$ and stride $s$ |

### D.2   TRANSFERABILITY OF AT-GAN

Another important issue for adversarial examples is the transferability across different models. To demonstrate the transferability of non-constrained adversarial examples, we use adversarial examples generated by attacking Model A (MNIST and Fashion-MNIST) and CNN (CelebA), to evaluate the attack success rates on Model C (MNIST and Fashion-MNIST) and VGG16 (CelebA). As shown in Table 7, non-constrained adversarial examples generated by AT-GAN exhibit moderate transferability.

Table 5: Architecture of WGAN_GP with auxiliary classifier for MNIST and Fashion-MNIST.

| Generator | Discriminator | Classifier |
|---|---|---|
| FC(1024) + Relu | Conv(64, $4 \times 4$) + LeakyRelu(0.2) | Conv(32, $3 \times 3$) + Relu |
| FC($7 \times 7 \times 128$) + Relu | Conv(128, $4 \times 4$) + LeakyRelu(0.2) | pooling(2, 2) |
| DeConv(64, $4 \times 4$) + Sigmoid | FC(1024) + LeakyRelu(0.2) | Conv(64, $3 \times 3$) + Relu |
| DeConv(1, $4 \times 4$) + Sigmoid | FC(1) + Sigmoid | pooling(2, 2) |
| FC(1024) | | |
| Dropout(0.4) | | |
| FC(10) + Softmax | | |

Table 6: Hyper-parameters of different attack methods on MNIST, Fashion-MNIST and CelebA.

| Attack | Datasets | | | Norm |
|---|---|---|---|---|
| | MNIST | Fashion-MNIST | CelebA | |
| FGSM | $\epsilon = 0.3$ | $\epsilon = 0.1$ | $\epsilon = 0.015$ | $\ell_\infty$ |
| PGD | $\epsilon = 0.3, \alpha = 0.075$, epochs = 20 | $\epsilon = 0.1, \alpha = 0.01$, epochs = 20 | $\epsilon = 0.015, \alpha = 0.005$, epochs = 20 | $\ell_\infty$ |
| R+FGSM | $\epsilon = 0.3, \alpha = 0.15$ | $\epsilon = 0.2, \alpha = 0.1$ | $\epsilon = 0.015, \alpha = 0.003$ | $\ell_\infty$ |
| Song's | $\lambda_1 = 100, \lambda_2 = 0$, epochs = 200 | $\lambda_1 = 100, \lambda_2 = 0$, epochs = 200 | $\lambda_1 = 100, \lambda_2 = 100$, epochs = 200 | N/A |
| AT-GAN | $\alpha = 2, \beta = 1$, epochs = 100 | $\alpha = 2, \beta = 1$, epochs = 100 | $\alpha = 3, \beta = 2$, epochs = 200 | N/A |

Table 7: Transferability of non-constrained adversarial examples and other search-based adversarial examples on three datasets. For MNIST and Fashion-MNIST, we attack Model C with adversarial examples generated on Model A. For CelebA dataset, we attack VGG16 using adversarial examples generated on CNN. Numbers represent the attack success rate (%).

| | MNIST | | | | Fashion-MNIST | | | | CelebA | | | |
|---|---|---|---|---|---|---|---|---|---|---|---|---|
| | Nor. | Adv. | Ens. | Iter. Adv. | Nor. | Adv. | Ens. | Iter. Adv. | Nor. | Adv. | Ens. | Iter. Adv. |
| FGSM | 46.7 | 4.2 | 1.7 | 4.6 | 68.9 | 23.1 | 20.8 | 14.8 | 15.6 | 4.3 | 3.3 | 4.1 |
| PGD | **97.5** | 6.5 | 4.1 | 4.1 | **84.7** | 27.6 | **39.6** | 14.6 | 18.3 | 4.3 | 3.1 | 4.1 |
| R+FGSM | 82.3 | 6.7 | 4.8 | 4.1 | 21.2 | 32.1 | 17.5 | 26.3 | 11.0 | 4.0 | 3.3 | 3.8 |
| Song's | 23.8 | 20.8 | 20.6 | **20.1** | 39.2 | **34.0** | 31.5 | **30.3** | 9.6 | **31.8** | 21.5 | **38.8** |
| AT-GAN | 65.3 | **24.6** | **27.9** | 17.2 | 58.0 | 22.7 | 32.0 | 15.2 | **63.7** | 15.4 | 16.5 | 17.6 |

## D.3 ABLATION STUDY

In this subsection, we investigate the impact of using different $\rho$ in the loss function. As $\rho$ could be constrained by both $\ell_0$ and $\ell_\infty$ norm, we test various bounds, using Model A on MNIST dataset, for $\rho$ in $\ell_0$ and $\ell_\infty$, respectively.

We first fix $\|\rho\|_\infty = 0.5$ and try various values for $\|\rho\|_0$, i.e. 0, 100, 200, 300, 400 (the maximum possible value is 784 for 28*28 input). The attack success rates are in Table 8. We can observe that different values of $\|\rho\|_0$ only have a little impact on the attack success rates, and the performances are very close for $\|\rho\|_0 = 0, 100, 200$. Figure 5 further illustrates some generated adversarial examples, among which we can see that there exist some slight differences on the examples. When $\|\rho\|_0 = 0$, AT-GAN tends to change the foreground (body) of the digits. When we increase the value of $\|\rho\|_0$ (100 and 200), AT-GAN is more likely to add tiny noise to the background and the crafted examples are more realistic to humans (for instance, smoother on digit 4). But if we continue to increase $\|\rho\|_0$ (300 or 400), AT-GAN tends to add more noise and the quality of the generated examples decays. To have a good tradeoff on attack performance and generation quality, we set $\|\rho\|_0 = 200$.

Table 8: Attack success rate (ASR, %) of AT-GAN with various values for $\|\rho\|_0$ using Model A on MNIST dataset.

| $\|\rho\|_0$ | 0 | 100 | 200 | 300 | 400 |
|---|---|---|---|---|---|
| ASR | 98.9 | 98.8 | 98.7 | 96.7 | 95.8 |

We then fix $\|\rho\|_0 = 200$ and test different values for $\|\rho\|_\infty$, i.e. 0, 0.1, 0.2, 0.3, 0.4, 0.5 (the maximum possible value is 1). The attack success rates are in Table 9. We can observe that different values of

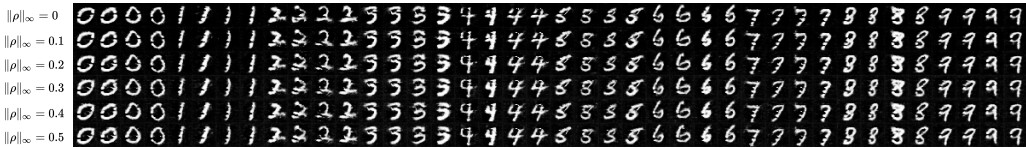

Figure 5: The adversarial examples generated by AT-GAN for various values of $\|p\|_0$.

$\|\rho\|_\infty$ have very little impact on the attack performance. Figure 6 further illustrates some generated adversarial examples, among which we can see that a little bit more noises are added for bigger $\|\rho\|_\infty$ but the differences are very tiny when $\|\rho\|_\infty = 0.2$ to $0.5$. So we simply set $\|\rho\|_\infty = 0.5$ in experiments, but other values of $\|\rho\|_\infty$ (0.2, 0.3, 0.4) also work.

Table 9: Attack success rate (ASR, %) of AT-GAN with various values for $\|\rho\|_\infty$ using Model A on MNIST dataset.

| $\|\rho\|_\infty$ | 0 | 0.1 | 0.2 | 0.3 | 0.4 | 0.5 |
|---|---|---|---|---|---|---|
| ASR | 98.9 | 99.2 | 98.9 | 98.9 | 98.9 | 98.7 |

Figure 6: The adversarial examples generated by AT-GAN for various values of $\|p\|_\infty$.

### D.4 HUMAN EVALUATION

To investigate the generating capacity of AT-GAN, we use the same input, and randomly pick 100 images for each category of MNIST generated by AT-GAN and the original generator, respectively. We then conduct human evaluation to determine whether each example is realistic. The evaluation results are in Table 10. We see that adversarial examples in some categories (e.g. 2, 4) are harder to be semantically meaningful than other categories (e.g. 0, 1). On average, however, the generating capability is close to that of the original generator.

Table 10: The evaluation results on the percentage of realistic images by human evaluation.

| Category | 0 | 1 | 2 | 3 | 4 | 5 | 6 | 7 | 8 | 9 | Average |
|---|---|---|---|---|---|---|---|---|---|---|---|
| Original | 100.0 | 100.0 | 93.0 | 94.0 | 98.0 | 96.0 | 99.0 | 100.0 | 98.0 | 100.0 | 97.8 |
| AT-GAN | 100.0 | 100.0 | 85.0 | 91.0 | 80.0 | 90.0 | 97.0 | 98.0 | 92.0 | 100.0 | 93.3 |

### D.5 AT-GAN ON CIFAR-10 DATASET

To further demonstrate the flexibility of AT-GAN, we implement AT-GAN on CIFAR-10 dataset using StyleGAN2-ada (Karras et al., 2020a), a recently proposed conditional GAN. The target classifier is wide ResNet w32-10 (Zagoruyko & Komodakis, 2016) by normal training (Nor.) and Iterative adversarial training (Iter.). The attack success rates are in Table 11. On normally trained models, PGD achieves the attack success rate of 100% while AT-GAN achieves the attack success rate of 93.5%. However, the adversarially trained model exhibits little robustness against AT-GAN and AT-GAN achieves attack success rate of 73.0%. In Figure 7, we illustrate some generated adversarial examples on CIFAR-10 dataset.

Table 11: Attack success rate (%) of adversarial examples generated by FGSM, PGD and AT-GAN against wide ResNet w32-10 by normal training (Nor.) and iterative adversarial training (Iter.).

| Model | FGSM | PGD | AT-GAN |
|---|---|---|---|
| Nor. | 92.3 | **100.0** | 93.5 |
| Iter. | 49.2 | 54.6 | 73.0 |

## D.6 AT-GAN on Target Attack

Here we show some non-constrained adversarial examples generated by AT-GAN for the target attack. The results are illustrated in Figure 8 for MNIST and Fashion-MNIST, and Figure 9 for CelebA. Instead of adding perturbations to the original images, AT-GAN transfers the generative model (GAN) so that the generated adversarial instances are not in the same shape of the initial examples (in diagonal) generated by the original generator. Note that for CelebA, the target adversarial attack is equivalent to the untarget adversarial attack as it is a binary classification task.

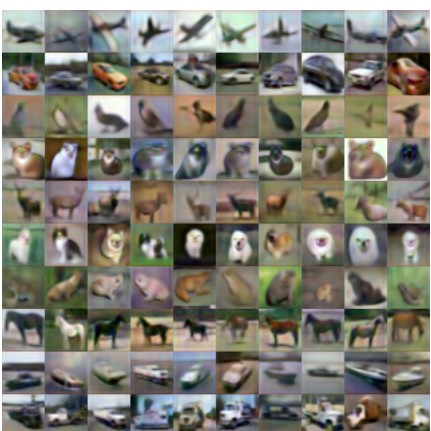

Figure 7: The adversarial examples generated by AT-GAN on CIFAR-10 dataset.

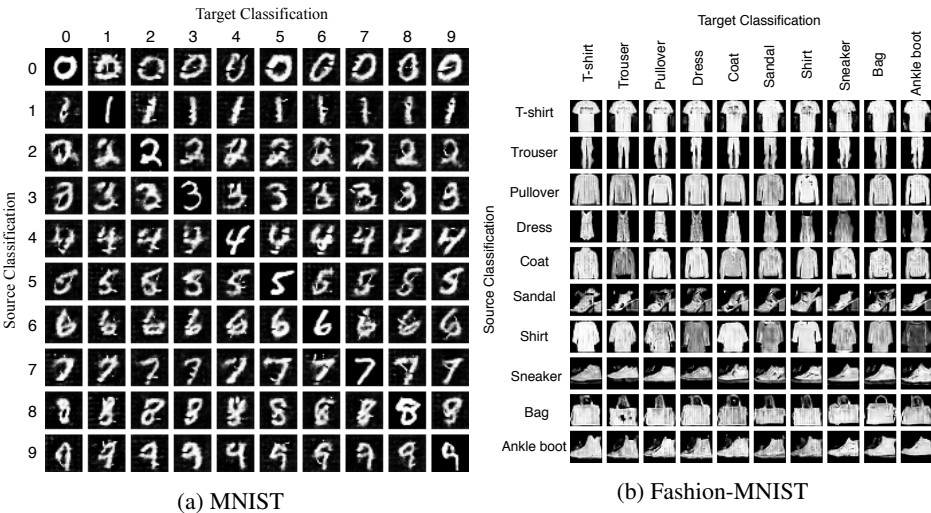

(a) MNIST

(b) Fashion-MNIST

Figure 8: Adversarial examples generated by AT-GAN to various targets with the same random noise input for each row. The images on the diagonal are generated by $G_{original}$ which are not adversarial examples and treated as the initial instances for AT-GAN.

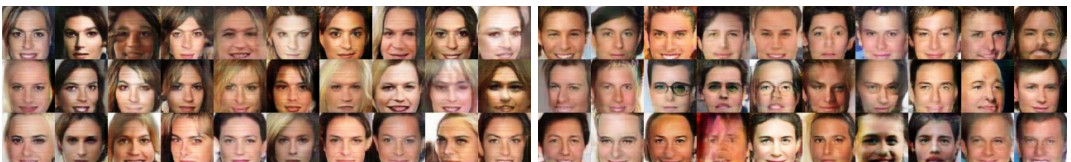

Figure 9: Adversarial examples generated by AT-GAN on CelebA dataset for the target attack.

## E  VISUALIZATIONS FOR THE ORIGINAL GAN AND AT-GAN

Here we provide some instances generated by the original GAN and AT-GAN with the same input noise and their difference on MNIST and Fashion-MNIST. The results are depicted in Figure 10 and 11. For different input noise, both the original GAN and AT-GAN output different instances. For each category with the same input noise, the difference between original GAN and AT-GAN is mainly related to the main content of image. For two different input noises, the differences between the original GAN and AT-GAN are not the same with each other, indicating that AT-GAN learns a distribution of adversarial examples different from the original GAN rather than just adds some universal perturbation vectors on the original GAN.

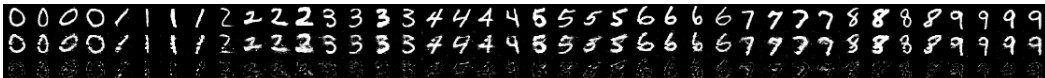

Figure 10: The instances generated by the original GAN and AT-GAN with the same input on MNIST. First row: the output of original GAN. Second row: the output of AT-GAN. Third row: The difference between the above two rows.

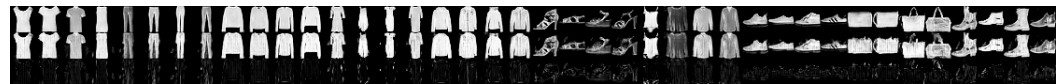

Figure 11: The instances generated by the original GAN and AT-GAN with the same input on Fashion-MNIST. First row: the output of original GAN. Second row: the output of AT-GAN. Third row: The difference between the above two rows.

