# OpenReview forum: "AT-GAN: An Adversarial Generative Model for Non-constrained Adversarial Examples"
_ICLR.cc/2021/Conference — Reject_

### Official Review · AnonReviewer4 · 2020-10-20
**AT-GAN: An Adversarial Generative Model for Non-constrained Adversarial Examples**

**Rating:** 5
**Confidence:** 4

**Review:**

This paper is to train a generative neural networks that can output adversarial examples. The main idea is to first train a normal GAN and then use the idea of transfer learning based on adversarial examples. The aim sounds good but the authors fail to clearly distinguish the idea with the exiting related methods theoretically or numerically. The idea of transferring is good (although not new), but after checking the implementation details, I have to say in the current version, the fact of transferring is quite limited.

Details:
+ the idea of generating adversarial examples by a trained GAN is interesting.
+ the writing is quite clear.
-  lack of comparison with existing related methods.
   Consider the core formulation, namely (2), which well describes the idea of this authors. But it is necessary to consider the following ideas:

   1). generating adversarial permutation (AdvGAN, AI-GAN):  min_G \| G(z,y) \|_p, s.t., f(z+G((z,y)) = y_t \neq y_s.
   It is to train the difference of G_original and G_attack and I think in the training aspects, this is almost equal to the proposed idea. The authors try to argue that the proposed model does not require an input. But in my opinion, no input is a disadvantage:  if only adversarial examples are needed, AdvGAN etc. can feed an random input to original GAN and then add perturbations; but if one wants to attack a specific image, the proposed method will fail.

   2). attack a GAN to generate adversarial examples (Song's): min_z' \|z - x\|, s.t., f(G(z,y)) \neq f(G(z'),y).
   The author may argue the Song's attack procedure takes longer time. However, the there is no training time additionally needed . Moreover, I guess the generating capability of Song's idea, which relies on the GAN and there are many well-designed ones, is better than the proposed one.  I would like to see the generating performance of the proposed method on more complicated datasets, e.g., on CIFAR or other HIGH-RESOLUTION images. Another good point of Song's idea is that almost all the attacks on images could be parallelly used. I do not know whether its ASR could be easily improved.

- The idea of transferring the original GAN to the attacking one is interesting. However, except of using the original GAN as the starting point, I cannot find other facts of "transferring". I would like to know if transferring learning technique could be used to reduce the number of required adversarial examples.

- The attack transferbility has not been tested. Since there is adversarial samples involved, the obtained GAN is expected to be related to the victim model.

Additional questions, mainly for the experiments' result
1. It is good that attack performance on adversarial trained NN is included. But where the adversarial examples come from? Are the examples are generated by AT-GAN?

2. How many examples and time are needed to train the AT-GAN?

3. Since the GAN has been changed, how about the generating capability, i.e., generating failure ratio of the AT-GAN should be reported.

---

> ### Author Response · Authors · 2020-11-25
> **Response to Review #4 (part 1/2)**
>
> Thank you for the valuable comments and suggestions. Below, we would like to address your main concerns:
>
> **General comments**: The main idea is to first train a normal GAN and then use the idea of transfer learning based on adversarial examples. The aim sounds good but the authors fail to clearly distinguish the idea with the exiting related methods theoretically or numerically. The idea of transferring is good (although not new), but after checking the implementation details, I have to say in the current version, the fact of transferring is quite limited.
>
> **A**: We try our best to clarify the differences of our work with existing related works, and address all your concerns in the following as well as in the revised paper. Note that we did not use any example for the transferring. We also add experiments following your suggestions, which could help improve our manuscript. Thank you.
>
> 1.	**Q**: comparison with existing related methods on generating adversarial perturbation (AdvGAN, AI-GAN).
>
>     **A**: All the perturbation-based adversarial attacks can be formulated as:
>     $$ min\|\delta\|_p, s.t., f(x+\delta) \neq y, $$
>     where $y$ is the true label of $x$. Both AdvGAN and AI-GAN aim to train a generator that can craft adversarial perturbation $\delta = G(x)$ ($x$ is an image):
>     $$ min_G |G(x)|_p, s.t., f(x+G(x)) = y_t \neq y,$$
>     where $y$ is its label and $y_t$ is target label. It is consistent to the goal of perturbation-based adversarial attacks.
>
>     In contrast, AT-GAN aims to generate adversarial examples through modeling the distribution of adversarial examples by transferring a pre-trained generator (z is a noise):
>     $$ min_G |G(z,y) –G_{ori}(z,y)|_p s.t. f( G(z,y) ) = y_t \neq y.$$
>
>     The differences are:
>
>     a) Different input: AdvGAN and AI-GAN take natural images as input while AT-GAN takes random noise as input.
>
>     b) Different output: AdvGAN and AI-GAN output the adversarial perturbation for the input image while AT-GAN outputs the adversarial example directly.
>
>     c) Different training procedure: AdvGAN is similar to train a normal GAN and AI-GAN also considers the adversarial examples for training, while AT-GAN transfers a pre-trained generator to model the distribution of adversarial examples and do not need adversaries for transferring.
>
>     We agree that our method could not generate adversarial perturbation for a natural image, but the goal of our method is different. We aim to learn the distribution of the adversaries so that the output looks like a natural image but misclassified by the target model. Under such scenario, we could generate diverse adversaries that are not limited to the natural image. AdvGAN and AI-GAN also could not generate adversarial examples directly as we did. Moreover, generating non-constrained adversarial examples is harder and might be very useful in some scenarios. For instance, it can help implement adversarial training to improve the model robustness in few-shot learning.
>
>     Surely you could first borrow a normal GAN to generate image and then use this image to add perturbation by any perturbation based methods, not only AdvGAN, AI-GAN, but also any gradient based methods like FGSM, PGD. But this is out of the scope of this discussion.
>
> 2.	**Q**: comparison with the work of Song's.
>
>     **A**: Song's method searches over the neighborhood of the input noise for the pre-trained AC-GAN in order to find a noise whose output image is misclassified by the target classifier. Their method is essentially based on search, while AT-GAN is trained as an adversarial generative model. The generating capability of both Song’s and ours rely on the GAN. We could also implement AT-GAN on other well-designed GANs for other datasets. Addressing your concern, we implement AT-GAN on CIFAR-10 dataset using StyleGAN2-ada (StyleGAN2 with adaptive discriminator augmentation) [1], a recently proposed conditional GAN. The target classifier is wide ResNet w32-10 [2] by normal training (Nor.) and Iterative training (Iter.). The attack success rates are as follows:
>
>     |   Model  |  PGD   |  FGSM  | AT-GAN |
>     |:--------:|:------:|:------:|:------:|
>     |  Nor.(%)   |  100.0 |  92.3  |  93.5  |
>     | Iter.(%)  |  54.6  |  49.2  |  73.0  |
>
>     On normally trained models, PGD achieves attack success rate of 100% while AT-GAN achieves attack success rate of 93.5%. However, the adversarially trained model exhibits little robustness against AT-GAN and AT-GAN achieves attack success rate of 73.0%. In Figure 5 in the Appendix D, we illustrate some generated adversarial examples on CIFAR-10 dataset. Thank you for the valuable suggestion.
>
>     [1] Tero Karras, Miika Aittala, Janne Hellsten, Samuli Laine, Jaakko Lehtinen, Timo Aila. Training Generative Adversarial Networks with Limited Data. NeurIPS 2020.
>
>     [2] Sergey Zagoruyko, Nikos Komodakis. Wide Residual Networks. BMVC 2016.

---

> > ### Author Response · Authors · 2020-11-25
> > **Response to Review #4 (part 2/2)**
> >
> > 3.	**Q**: I would like to know if transfer learning technique could be used to reduce the number of required adversarial examples.
> >
> >     **A**: AT-GAN transfers the conditional generator which can craft benign examples to generate adversarial examples. But different from transfer learning, the aim of AT-GAN is to generate the examples that fool the target classifier guaranteed by Eq. (3)
> >
> >     $$L_1 = \mathbb{E}_{z\sim p_z}[H(f(G_{attack}(z,y_s)), y_t)],$$
> >
> >     and are realistic to humans guaranteed by Eq. (4)
> >
> >     $$L_2 = \mathbb{E}_{z\sim p_z}[\|G_{original}(z,y_s) + \rho - G_{attack}(z,y_s)\|]_p.$$
> >
> >     With the two loss functions, AT-GAN is trained to transfer the generator $G_{original}$ that models the distribution of benign examples to $G_{attack}$ that models the distribution of adversarial examples. This process is different from the training process of GANs and we do not need any adversarial examples for the transferring process.
> >
> > 4.	**Q**: The attack transferability.
> >
> >     **A**: We add experiments and use adversarial examples generated on Model A to attack Model C for MNIST dataset. The results are depicted as follows:
> >
> >     | | Nor. | Adv. | Ens. | Iter. |
> >     |-|------|------|------|-------|
> >     | FGSM | 46.7 | 4.2| 1.7| 4.6
> >     | PGD | **97.5**| 6.5| 4.1| 4.1
> >     |R+FGSM| 82.3 | 6.7| 4.8| 4.1
> >     |Song's| 23.8| 20.8 | 20.6 | **20.1**
> >     |AT-GAN| 65.3| **24.6** | **27.9**| 17.2
> >
> >     We can see that the examples generated by AT-GAN exhibit moderate transferability.
> >
> > 5.	**Q**: The adversarial examples used for adversarial training.
> >
> >     **A**: Adversarial training aims to defend various adversarial attacks but not limited to the adopted attack for adversarial training. Therefore, the examples we used are not generated by AT-GAN. We adopt three adversarial training in our experiments: a) adversarial training (Adv.) [1] uses adversarial examples generated by FGSM, b) ensemble adversarial training (Ens.) [2] uses adversarial examples generated by R+FGSM on the ensemble of models, c) Iterative adversarial training (Iter.) [3] uses adversarial examples generated by PGD.
> >
> >     [1] Ian Goodfellow, Jonathon Shlens, Christian Szegedy. Explaining and Harnessing Adversarial Examples. ICLR 2015.
> >
> >     [2] Florian Tramèr, Alexey Kurakin, Nicolas Papernot, Ian Goodfellow, Dan Boneh, Patrick McDaniel. Ensemble Adversarial Training: Attacks and Defenses. ICLR 2018.
> >
> >     [3] Aleksander Madry, Aleksandar Makelov, Ludwig Schmidt, Dimitris Tsipras, Adrian Vladu. Towards Deep Learning Models Resistant to Adversarial Attacks. ICLR 2018.
> >
> > 6.	**Q**: Number of examples and time needed to train AT-GAN.
> >
> >     **A**: As in A3, we do not need any adversarial examples for the transferring process. For the training time, it takes about 8 minutes for transferring the generator of AT-GAN for Model A on MNIST. As we can craft numerous adversarial examples directly once the generator is transferred, we do not consider such time in the comparison for crafting 1,000 examples in the experiments. We have clarified it in the revision, thank you.
> >
> > 7.	**Q**: The generating capability, i.e., generating failure ratio, of AT-GAN.
> >
> >     **A**: We use the same input, and randomly pick 100 images for each category of MNIST generated by AT-GAN and the original generator, respectively. We then conduct human evaluation to determine whether each example is realistic. The evaluation results on the percentage of realistic images are as follows:
> >
> >     |   Category  |  0  |  1  |  2  |  3  |  4  |  5  |  6  |  7  |  8  |  9  |  Average |
> >     |-------------|-----|-----|-----|-----|-----|-----|-----|-----|-----|-----|----------|
> >     | Original(%)    | 100 | 100 | 93  | 94  | 98  |  96 | 99  | 100 | 98  | 100 |    97.8  |
> >     | AT-GAN(%)   | 100 | 100 | 85  | 91  | 80  |  90 | 97  | 98  | 92  | 100 |    93.3  |
> >
> >     We see that adversarial examples in some categories (e.g. 2, 4) are harder to be semantically meaningful than other categories (e.g. 0, 1). On average, however, the generating capability is close to that of the original generator.
> >
> >     We have added the human evaluation in the revision. Thank you.

---

### Official Review · AnonReviewer2 · 2020-10-25
**A meaningful solution to find non-constrained adversarial examples by using adversarial transfer on generative adversarial net**

**Rating:** 7
**Confidence:** 3

**Review:**

This paper proposed the adversarial transfer on generative adversarial net (AT-GAN) to train an adversarial generative model that can directly produce adversarial examples. In the other way, AT-GAN could generate the adversarial examples directly for any input noise. Such a generative model was able to draw non-constrained adversarial examples.

Pros:
This paper is clearly written with reasonable paper organization covering background, model design, mathematical formula and experiments. The goal of this work is obvious with experimental justification. Mathematical description and experimental illustration are desirable to show the merit of this method.

Cons:
The reasons of using AC-GAN and WGAN-GP as the pre-train stage are missing.

---

> ### Author Response · Authors · 2020-11-25
> **Response to Review #2**
>
> We appreciate the positive remarks that greatly encourage us, and the valuable suggestion made by the reviewer that have helped to improve the quality of our paper in the revised version.
>
> 1.	**Q**: The reasons of using AC-GAN and WGAN-GP as the pre-training stage.
>
>     **A**: There are two main reasons for adopting AC-GAN and WGAN-GP in the pre-training stage for our AT-GAN implementation. 1) In the literature, the combination of AC-GAN and WGAN-GP could build a powerful generative model and can craft realistic images on the evaluated datasets. 2) Song et al. [1] also utilize the same combination, and we follow their experimental setting for the fair comparison.
>
>     But AT-GAN is not limited to the above GANs. Actually, all conditional GANs that can craft realistic examples could be used for the implementation of AT-GAN in the pre-training stage. For instance, we add experiments on CIFAR-10 using StyleGAN2-ada (StyleGAN2 with adaptive discriminator augmentation) [2], and illustrate some generated examples in Appendix D.3. We have clarified it in the revision.
>
>     [1] Yang Song, Rui Shu, Nate Kushman, Stefano Ermon. Constructing Unrestricted Adversarial Examples with Generative Models. NeurIPS 2018.
>
>     [2] Tero Karras, Miika Aittala, Janne Hellsten, Samuli Laine, Jaakko Lehtinen, Timo Aila. Training Generative Adversarial Networks with Limited Data. NeurIPS 2020.

---

### Official Review · AnonReviewer1 · 2020-10-28
**a straightforward idea**

**Rating:** 6
**Confidence:** 4

**Review:**

The paper proposes AT-GAN (Adversarial Transfer on Generative Adversarial Net) to train an adversarial generative model that can directly produce adversarial examples. Different from previous works, the study aims to learn the distribution of adversarial examples so as to generate semantically meaningful adversaries. AT-GAN achieves this goal by ﬁrst learning a generative model for real data, followed by transfer learning to obtain the desired generative model. Once trained and transferred, AT-GAN could generate adversarial examples directly for any input noise, denoted as non-constrained adversarial examples. Some experiments and visualizations show that AT-GAN can generate some diverse adversarial examples that are realistic to human perception, and yields higher attack success rates against adversarially trained models.

Overall, the idea seems straightforward. Benefiting from the GAN, the proposed model could learn the distribution of adversarial examples to attach the target models. The paper is clearly written and some experiments are conducted. However, I have some concerns as below:

1. In the loss function, $\rho$ controls the difference between the outputs of the original and attach GANs, it is expected to see the performance and generated examples with different $\rho$.

2. The idea seems incremental. The main contribution is to transfer a pre-trained GAN to attach GAN to fool the classifiers. The novelty could be further summarized by highlighting the difference with most related works including but not limited to the aforementioned ones. The current manuscript makes the work seem like a straightforward combination of many existing approaches.

3. Some experiment settings are not clear. A brief introduction to Model A to B should be given in the main paper, though the details is provided in Appendix.

As most concerns of mine are addressed by the rebuttal and I would like to rise my score.

---

> ### Author Response · Authors · 2020-11-25
> **Response to Review #1 (part 1/2)**
>
> We appreciate the reviewer’s constructive suggestions and have performed the corresponding revisions. Below, we would like to address your main concerns.
>
> 1.	**Q**: It is expected to see the performance and generated examples with different $\rho$.
>
>     **A**: We add experiments to investigate the impact of using different $\rho$ in the loss function. As $\rho$ could be constrained by both $\ell_0$ and $\ell_\infty$ norm, we test various bounds, using Model A on MNIST dataset, for $\rho$ in $\ell_0$ and $\ell_\infty$, respectively.
>
>     a) We first fix $\|\rho\|_\infty=0.5$ and try various values for $\|\rho\|_0$, i.e. 0, 100, 200, 300, 400 (the maximum possible value is 784 for 28$\times$28 input). The attack success rates (ASR) are as follows:
>
>    $\|\rho\|_0$|0|100|200|300|400
>     -|-|-|-|-|-
>     ASR(%)|98.9|98.8|98.7|96.7|95.8
>
>     We can observe that different values of $\|\rho\|_0$ only have a little impact on the attack success rates, and the performances are very close for $\|\rho\|_0$ = 0, 100, 200. Figure 6 in Appendix D further illustrates some generated adversarial examples, among which we can see there exist some slight differences on the examples. When $\|\rho\|_0=0$, AT-GAN tends to change the foreground (body) of the digits. When we increase the value of $\|\rho\|_0$ (100 and 200), AT-GAN is more likely to add tiny noise to the background and the crafted examples are more realistic to humans (for instance, smoother on digit 4). But if we continue to increase $\|\rho\|_0$ (300 or 400), AT-GAN tends to add more noise and the quality of the generated examples decays. To have a good tradeoff on attack performance and generation quality, we set $\|\rho\|_0=200$.
>
>     b)	We then fix $\|\rho\|_0=200$ and test different values for $\|\rho\|_\infty$, i.e. 0, 0.1, 0.2, 0.3, 0.4, 0.5 (the maximum possible value is 1). The attack success rates (ASR) are as follows:
>
>     $\|\rho\|_\infty$|0|0.1|0.2|0.3|0.4|0.5
>     -|-|-|-|-|-|-
>     ASR(%)|98.9|99.2|98.9|98.9|98.9|98.7
>
>     We can observe that different values of $\|\rho\|_\infty$ have very little impact on the attack performance. Figure 7 in Appendix D further illustrates some generated adversarial examples, among which we can see that a little bit more noises are added for bigger $\|\rho\|_\infty$ but the differences are very tiny when $\|\rho\|_\infty = 0.2$ to 0.5. So we simply set $\|\rho\|_\infty=0.5$ in experiments, but other values of $\|\rho\|_\infty$ (0.2, 0.3, 0.4) also work.
>
>     We have added it in the revision. Thank you again for the valuable suggestion!

---

> > ### Author Response · Authors · 2020-11-25
> > **Response to Review #1 (part 2/2)**
> >
> > 2.	**Q**: The idea seems incremental. The novelty could be further summarized by highlighting the difference with most related works including but not limited to the aforementioned ones.
> >
> >     **A**: AT-GAN aims to learn the distribution of adversarial examples so as to generate semantically meaningful adversaries by transferring a pre-trained GAN, and there are no adversarial examples involved during the training. Once transferred, AT-GAN can directly generate adversarial examples from any input noise.
> >
> >     Here we highlight the differences with most related works as follows:
> >
> >     + **NAG, AdvGAN and AI-GAN vs. AT-GAN.** NAG [1], AdvGAN [2] and AI-GAN [3] focus on crafting adversarial perturbations by GANs. NAG [1] takes random noise as input and crafts image-agnostic adversarial perturbation. Such perturbation can be added to many natural images to craft the adversaries. AdvGAN [2] and AI-GAN [3] both use natural images as inputs, and generate the corresponding adversarial perturbations using GAN for the input image. AdvGAN fixes the target class for the generation, while AI-GAN uses projected gradient descent (PGD) attack to inspire the training of GAN and the target class is used as an input. In contrast, AT-GAN does not use any natural image as the input, but generates the adversaries directly from any random noise. Further, compared with AI-GAN, we do not use adversarial examples for the training.
> >
> >     + **Song's vs. AT-GAN.** Song's method [4] searches over the neighborhood of the input noise for the pre-trained AC-GAN in order to find a noise whose output image is misclassified by the target classifier. They define such adversaries as the unrestricted adversarial examples, however, their adversaries are still constrained by the original input noise. Their method is essentially based on search, while AT-GAN is trained as an adversarial generative model and our output is not constrained by any neighborhood.
> >
> >     + **PS-GAN vs. AT-GAN** PS-GAN [5] pre-processes an input seed patch (a meaningful small image) to adversarial patch that will be added to a natural image (such as a traffic sign) to craft an adversarial example, and an attention model is used to locate the attack area on the natural image. Their method uses GAN to generate meaningful adversarial patch based on the original patch, and paste that patch on a natural image. Though GAN is involved, their task is very different from ours.
> >
> >     In summary, existing works either are based on a search in a neighborhood of the input, or use a generative model to generate the perturbations or patches which will then be added to a natural image. Differs to existing works, we aim to model the distribution of adversarial examples by transferring a pre-trained GAN and generate non-constrained adversarial examples directly and quickly from any random noise.
> >
> >     We have highlighted the differences and make it clearer in the revision.
> >
> >     [1] Konda Reddy Mopuri, Utkarsh Ojha, Utsav Garg, R. Venkatesh Babu. NAG: Network for Adversary Generation. CVPR 2018.
> >
> >     [2] Chaowei Xiao, Bo Li, Jun-Yan Zhu, Warren He, Mingyan Liu, Dawn Song. Generating Adversarial Examples with Adversarial Networks. IJCAI 2018.
> >
> >     [3] Tao Bai, Jun Zhao, Jinlin Zhu, Shoudong Han, Jiefeng Chen, Bo Li. AI-GAN: Attack-Inspired Generation of Adversarial Examples. arXiv Preprint arXiv:2002.02196, 2020.
> >
> >     [4] Yang Song, Rui Shu, Nate Kushman, Stefano Ermon. Constructing Unrestricted Adversarial Examples with Generative Models. NeurIPS 2018.
> >
> >     [5] Aishan Liu, Xianglong Liu, Jiaxin Fan, Yuqing Ma, Anlan Zhang, Huiyuan Xie, Dacheng Tao. Perceptual-Sensitive GAN for Generating Adversarial Patches. AAAI 2019.
> >
> > 3.	**Q**: Some experiment settings are not clear. A brief introduction to Model A to B should be given in the main paper, though the details is provided in Appendix.
> >
> >     **A**:  We make the experiment settings clearer, and add a brief introduction of Model A to D to the main paper in the revision. Thank you again for the clarity check.

---

### Decision · Program_Chairs · 2021-01-07
**Final Decision**

**Decision:**

Reject

**Comment:**

I thank the authors and reviewers for their discussions about this paper. The proposed AT-GAN is a GAN-based method to generate adversarial examples. Similar methods (e.g. Song et al) have been proposed to use GANs to generate adv. examples more efficiently. Authors show their method has some numerical benefits. However, more experiments are needed to further justify it. Also, creating "unrestrictive" adv. examples can cause a risk of generating samples where the true label is flipped. Authors need to clarify it. Given all, I think the paper needs a bit of more work to be accepted. I recommend authors to address the aforementioned concerns in the updated draft.

-AC